# Production of Bacterial Exopolysaccharides: Xanthan and Bacterial Cellulose

**DOI:** 10.3390/ijms241914608

**Published:** 2023-09-27

**Authors:** Viktor V. Revin, Elena V. Liyaskina, Marina V. Parchaykina, Irina V. Kurgaeva, Kristina V. Efremova, Nikolai V. Novokuptsev

**Affiliations:** Department of Biotechnology, Biochemistry and Bioengineering, National Research Ogarev Mordovia State University, 430005 Saransk, Russia; liyaskina@yandex.ru (E.V.L.); mary.isakina@yandex.ru (M.V.P.); irina.kurgaewa@yandex.ru (I.V.K.); kristina.efremova.1221@mail.ru (K.V.E.); nikolay.novokuptsev@yandex.ru (N.V.N.)

**Keywords:** biopolymers, biomacromolecules, bacterial exopolysaccharides, xanthan, bacterial cellulose, functional materials

## Abstract

Recently, degradable biopolymers have become increasingly important as potential environmentally friendly biomaterials, providing a wide range of applications in various fields. Bacterial exopolysaccharides (EPSs) are biomacromolecules, which due to their unique properties have found applications in biomedicine, foodstuff, textiles, cosmetics, petroleum, pharmaceuticals, nanoelectronics, and environmental remediation. One of the important commercial polysaccharides produced on an industrial scale is xanthan. In recent years, the range of its application has expanded significantly. Bacterial cellulose (BC) is another unique EPS with a rapidly increasing range of applications. Due to the great prospects for their practical application, the development of their highly efficient production remains an important task. The present review summarizes the strategies for the cost-effective production of such important biomacromolecules as xanthan and BC and demonstrates for the first time common approaches to their efficient production and to obtaining new functional materials for a wide range of applications, including wound healing, drug delivery, tissue engineering, environmental remediation, nanoelectronics, and 3D bioprinting. In the end, we discuss present limitations of xanthan and BC production and the line of future research.

## 1. Introduction

Bacterial exopolysaccharides (EPSs) are valuable extracellular eco-friendly biopolymers used in various fields of science, industry, medicine, and technology due to their biocompatibility, nontoxicity, biodegradability, and functional characteristics. Recently, several reviews have provided a comprehensive overview of the fundamentals of bacterial EPSs, including their classification, source, properties, biosynthetic pathways, functions in the microbial community, and applications [1,2,3,4,5,6,7,8,9]. EPSs are cost-effective alternatives to plant and animal-derived polysaccharides because bacteria can produce them in large quantities by biotechnological processes using low-cost substrates such as industrial wastes in a short time regardless of the season and climate. They exhibit the presence of a great number of functional groups (hydroxyl, carboxyl, carbonyl, acetate, etc.), which enable the modification of their molecules using chemical and physical techniques to obtain composites and materials with improved functional properties [5,10,11,12].

EPS-producing bacteria are ubiquitous and can be isolated from aquatic and terrestrial environments, such as marine water, wastewater, soils, plants, fruits, vegetables, gut microbiome, and fermented food [2,13]. For example, many BC-producing strains were isolated from the kombucha community [14,15,16,17,18], vinegar [19], fruit, and fruit juices [20,21,22]. Revin and Liyaskina et al. obtained the highly productive strain *Paenibacillus polymyxa* 2020, first isolated from wasp honeycombs [23]. A recent review by Netrusov et al. (2023) summarized the current research progress on BC, xanthan, and levan-producing bacterial strains, including their characteristics and isolation sources [2]. The review by Ibrahim et al. (2022) focuses on the exopolysaccharides obtained from several extremophilic microorganisms, their synthesis, and manufacturing optimization for better cost and productivity [13]. Recently, several reviews on EPSs produced by lactic acid bacteria have been reported [24,25,26,27]. For example, the review by Jurášková et al. (2022) discussed and summarized the latest advances on the biosynthesis, structure, and properties of EPSs derived from *Lactobacillus*,* Leuconostoc*,*Streptococcus*, Lactococcus, Lactiplantibacillus, Limosilactobacillus, and Weissella genera [24]. EPSs play an important role in bacterial physiology and ecology. They protect cells against extreme temperature, unfavorable pH values, osmotic stress, salinity, aridity, UV-rays, phagocytosis, and chemical agents, and play an important role in bacterial aggregation, adhesion, and biofilm formation [28,29,30,31,32,33,34]. Some of the most-used EPSs are alginate from the *Azotobacter*, *Pseudomonas* genera, xanthan from *Xanthomonas* sp., dextran from the *Leuconostoc*, *Lactobacillus*, *Streptococcus* genera, curdlan from *Alcaligenes faecalis*, *Rhizobium radiobacter*, *Agrobacterium* sp., gellan from the *Sphingomonas* and *Pseudomonas* genera, hyaluronan from *Streptococcus* sp., levan from *Bacillus* sp., *Paenibacillus* sp., *Halomonas* sp., *Zymomonas* sp., BC from *Komagataeibacter* sp., and others [2]. In recent years, a large number of new strains of EPS-producing bacteria have been isolated, among them the Gram-negative bacteria *Komagataeibacter* sp. and *Xanthomonas* sp., the main producers of BC and xanthan, which deserve special attention since their metabolism is well studied and many complete genome sequences were obtained, which is the basis for obtaining highly productive strains using genetic and metabolic engineering [2].

Bacterial EPSs are classified into two types: homopolysaccharides, which are either unbranched or branched and composed of a single type of monosaccharides such as D-glucose, D-fructose, or D-galactose linked through glycosidic bonds; and heteropolysaccharides, which contain two or more units of different sugars, such as pentose (D-ribose, D-arabinose, D-xylose), hexose (D-glucose, D-galactose, D-mannose), N-acetylated monosaccharides (N-acetyl-glucosamine and N-acetyl-galactosamine), or uronic acids (D-glucuronic acid, D-galacturonic acid), and may be branched or unbranched [6,7,24]. There are four general mechanisms for bacterial EPS biosynthesis: the Wzx/Wzy-dependent pathway, where individual repeating units are assembled by several glycosyltransferases, the ATP-binding cassette (ABC) transporter-dependent pathway, the synthase-dependent pathway, and extracellular biosynthesis by glucan sucrases [1,7,9]. Homopolysaccharides are commonly synthesized using synthase and extracellular synthesis pathways, while heteropolysaccharides are produced by the Wzx/Wzy-dependent pathway and the ABC transporter-dependent pathway. For example, heteropolysaccharide xanthan is synthesized through a Wzy-dependent pathway and homopolysaccharide BC through the synthase pathway [35,36,37]. The biosynthesis of xanthan and BC begins with the synthesis of exopolysaccharide precursors UDP-glucose for BC and UDP-glucose, GDP-mannose, and UDP-glucuronic acid for xanthan. Bacterial EPSs have many unique beneficial properties such as biocompatibility, biodegradability, non-toxicity, a high degree of polymerization, the ability for gelation, high adhesive ability, viscoelasticity, pseudo-plasticity, a thixotropic nature, renewable sourcing, and easy modification. In addition, some bacterial EPSs also have extensive bioactivities, including antibacterial, antifungal, antiviral, antioxidant, anti-inflammatory, antitumor, antidiabetic, antiulcer, anticoagulant, immunomodulatory, prebiotic, wound healing, and cholesterol-lowering activities [1,3,25,27,35,36,37,38,39,40,41,42,43]. Therefore, they have extensive commercial applications in biomedicine, food, pharmaceuticals, cosmetics, electronics, environmental remediation, and the oil and gas industries [1,2,3,4,5,6,7,8,9]. Among bacterial EPSs, the homopolysaccharide BC and the heteropolysaccharide xanthan rank high. They exhibit many unique properties. For example, BC is a 3D nanostructured material with high crystallinity and a large surface area, and xanthan, along with other valuable properties, has pseudo-plasticity, a thixotropic nature, and is resistant to various environmental factors [2]. Over the last years, there have been obtained a great number of xanthan and BC-based biocomposite materials with additional valuable properties, including antimicrobial activity, antioxidant activity, electromagnetic properties, catalytic activity, and others [5,10,11,12].

In recent years, such EPSs as BC and xanthan have attracted special attention. One of the main exopolysaccharides produced nowadays is xanthan, with a production of 50,000 tons per year [44]. The global xanthan market has increased significantly (the rate being 5.6%) since 2019. Moreover, its market value is expected to reach USD 1.2 billion by 2030 [9]. Its demand increase is due to applications in the food, agrochemical, cosmetics, driller fluid, and foam stabilizer segments. In addition, in recent years, the range of its application has expanded significantly. Over the past few decades, BC production has also exponentially increased. The BC market was valued at USD 207.36 million in 2016 and is expected to surpass a valuation of USD 700 million in 2026 [45]. Xanthan is one of the most expensive EPSs due to the use of only glucose and sucrose as carbon sources and the cost of the downstream process (approximately 50% of the final cost), since a high purity level is required when it is used in the food industry [9]. BC is also an expensive EPS due to the use of glucose, fructose, and sucrose as carbon sources and low productivity strains (usually not more than 10 g/L) [46].

Therefore, the aim of this review is to summarize the strategies for the cost-effective production of such important industrial exopolysaccharides as xanthan and BC. By the example of these two bacterial EPSs, which differ in their chemical structure and properties, we aim at demonstrating common approaches to their efficient production and obtaining new functional materials based on them for a wide range of applications. Section 1 presents a general idea of the subject highlighting the specific features of xanthan and cellulose compared to other bacterial EPSs. Section 2 provides information on properties, biosynthesis, as well as BC and xanthan producers. Section 3 summarizes strategies for the cost-effective production of xanthan and BC, including their production from wastes, EPS-producing bacteria co-cultivation, biocatalytic technologies, and genetic and metabolic engineering. Section 4 introduces the recent advances in obtaining new functional materials for a wide range of applications, including wound healing, drug delivery, tissue engineering, environmental remediation, nanoelectronics, and 3D bioprinting. At the end of the review, we recommend further studies and investigation for highly efficient BC and xanthan production and for obtaining new functional materials based on them.

## 2. Properties and Biosynthesis of BC and Xanthan

Xanthan was one of the first bacterial EPSs utilized for industrial production. Compared to other microbial polysaccharides, it is cost-competitive, and therefore the best option both in terms of performance and economically [47]. The global xanthan market is estimated to experience a 15% increase by 2027, which will result in about USD 455.9 million [48]. The main xanthan gum manufacturers are Jungbunzlauer, ADM, Cargill, CP Kelco, Deosen Biochemicals, Fufeng Group, IFF (Dupont), and Meihua Group [49]. The global BC market was valued at USD 250 million in 2017 and estimated to reach USD 680 million by the end of 2025 [7]. The BC market price is about USD 25/kg for the packaged final product [45]. The market price of xanthan is of the next lower order and amounts to about USD 1500–4000/ton [47].

The BC and xanthan chemical structures are comparable. They have cellobiose as a repeating unit. BC is a linear homopolysaccharide with D-glucose residues interconnected by β-1,4-glycosidic bonds (Figure 1A). A cellulose macromolecule is composed of thousands of glucose residues. BC is characterized by a high degree of polymerization, which ranges from 16,000 to 20,000, while for a plant-derived cellulose it is approximately 13,000 [50]. The molecular weight of BC is approximately 2300 kDa in static culture and slightly lower in 10 L and 50 L bubble column bioreactor cultivation, about 1800 and 1700 kDa, respectively [51]. The order in which cellulose macromolecules are arranged is maintained mainly due to the forces of intramolecular and intermolecular hydrogen bonds. In the cellulose structure, each glucose unit is presented with three hydroxyl groups, hydrogen bonds being very important. They have an impact on BC physical, physicochemical, and chemical properties and provide the fibers with high mechanical strength and insolubility in most solvents. In spite of the similar chemical composition, BC structure and properties differ sufficiently from those of plant cellulose [52,53]. Bacteria produce extracellular biodegradable and completely non-toxic cellulose with high purity. BC molecules are arranged strictly parallel to each other to form crystalline microfibrils 100 times thinner than plant cellulose microfibrils. This unique three-dimensional network structure of BC is responsible for most of its properties, such as high tensile strength, high degrees of polymerization and crystallinity (up to 90%), superior mechanical properties (Young’s modulus about 15–35 GPa and tensile strength of 200–300 MPa), a large surface area (>150 m^2^/g), high elasticity, and water retention [37,54,55,56,57,58].

Xanthan is a heteropolysaccharide containing a cellulose-like backbone of β-1,4-linked glucose units substituted alternately with a trisaccharide side chain composed of two mannose units separated by a glucuronic acid. The internal mannose is mostly O-acetylated, and the terminal mannose can be substituted by a pyruvic acid residue (Figure 1B). Due to the presenting glucuronic and pyruvic acid in the side chain, xanthan represents a highly charged polysaccharide with a very rigid polymer backbone. The content of pyruvate in xanthan ranges from 2.5–4.4%; this suggests that not every residue of the terminal D-mannopyranose in the side chain carries a pyruvate ketal group. Wu M. et al. obtained a genetically engineered *X. campestris* strain CGMCC 15155, which produces high-viscosity xanthan with a pyruvate content of 8.69% [59]. The low content of the pyruvyl group decreases the viscosity, while the high pyruvyl content contributes to the gel viscosity [60]. The ratio of acetate in the xanthan molecule can also vary depending on the polymer sample. Higher acetyl content decreases the gelling capacity of xanthan gum in an aqueous solution [60]. In addition, the hydrogen of the acetyl, pyruvic, and carboxyl groups in the D-glucuronic acid residue can be replaced by any cation. Thus, the pyranose sugar blocks in xanthan are not always structurally identical to each other, and the present acetic and pyruvic acids form an anionic polysaccharide. Xanthan has a high molecular mass of about 2 × 10^6^ to 2 × 10^7^ Da, which is influenced both by bacterium strains and fermentation conditions [48]. The conformation of a polysaccharide macromolecule changes differently depending on pH, temperature, ionic strength, fermentation duration, medium composition, production method, etc. [61]. Xanthan, in contrast to BC, is a water-soluble bacterial EPS and has high solubility in both cold and hot water. Xanthan molecules in aqueous solutions are prone to self-association, and a gel forms with an increase in the ionic strength of the solution or the concentration of the polysaccharide. It is a three-dimensional network formed from double helixes of xanthan linked by intermolecular hydrogen bonds [62]. Xanthan has a pseudoplastic nature; that is, the viscosity inversely changes with the shear rate of a xanthan solution. Xanthan molecular structure and conformational state are closely associated with its rheology, stability, and function [63,64]. These properties enable it to be used as a thickening, dispersant, emulsifier, and viscous aqueous solution at low concentrations (0.05–1%). Xanthan has a high tolerance to deviations in the pH range of 2–12 and a high resistance to temperature changes. These properties confer industrial relevance and can explain the wide commercial EPS acceptance [65]. The commercial demand is a key point stimulating studies to increase xanthan production on an industrial scale by sustainable processes to exploit the microorganism potential [66,67].

Xanthan is produced by bacteria of the genus *Xanthomonas* belonging to the class Gammaproteobacteria of the phylum Proteobacteria. The strain type is *X. campestris* ATCC 33913T [2]. The *Xanthomonas* spp. differentiates further into pathovars depending on the host plant [68]. Although *X. campestris* is most commonly employed for the industrial production of xanthan, there are several other strains of genus *Xanthomonas* which can produce xanthan, including *X. pelargonii*, *X. phaseoli*, *X. malvacearum*, *X. arbicola*, *X. axonopodis*, and *X. citri* [2,69,70,71]. BC, unlike xanthan, can be produced by bacteria of different genera, such as Gram-negative bacteria of the genera *Komagataeibacter* (*Gluconacetobacter*) [14,72], *Gluconobacter* [73], *Acetobacter* [74], *Achromobacter* [75], *Agrobacterium* [76], *Enterobacter* [77], *Pseudomonas* [78], *Rhizobium* [79], *Salmonella* [80], and others, as well as Gram-positive bacteria of the genera *Bacillus* [81], *Sarcina*, and *Rhodococcus* [82]. The most common and highly productive BC producers are acetic bacteria species of the *Komagataeibacter* genus belonging to the *Acetobacteraceae* family, class Alphaproteobacteria, phylum Proteobacteria [2].
Figure 1Schematic overview of BC (**A**) and xanthan (**B**) chemical structure and biosynthesis machinery of BC (**C**) and xanthan (**D**). Adapted from Refs. [36,83] (open access).
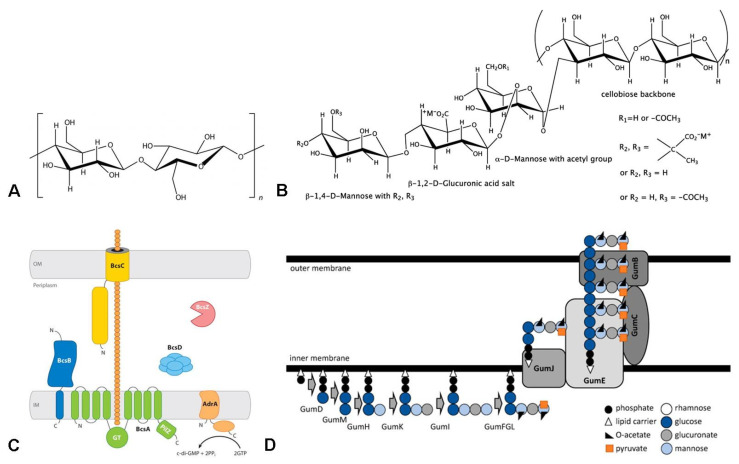



BC producers *Komagataeibacter* sp. and xanthan producers *Xanthomonas* sp. are Gram-negative bacteria that have the periplasmic space between the bacterial cytoplasmic and outer membranes, where important steps in EPS biosynthesis occur (Figure 1C,D). The biosynthesis of BC and xanthan begins intracellularly with the synthesis of exopolysaccharide precursors uridine diphosphate glucose (UDP)-glucose for BC and UDP-glucose, GDP-mannose, and UDP-glucuronic acid for xanthan. BC biosynthesis involves three stages, including UDP-glucose synthesis through a series of enzymatic reactions, cellulose molecular chain synthesis under the function of cellulose synthase (CS), and cellulose crystallization and polymerization. In Gram-negative bacteria, CS is composed of four polypeptide subunits, including a catalytic BcsA subunit, an inner membrane-anchored BcsB subunit having a carbohydrate-binding domain, a BcsC subunit (secretion of a glucan chain through the outer membrane), and a BcsD subunit (crystallization subunit) (Figure 1C) [83]. One of the well-characterized mechanisms regulating BC biosynthesis is the allosteric activation of BcsA with a cyclic di-GMP (c-di-GMP) molecule, a universal bacterial second messenger discovered by Moshe Benziman and his group in 1987 after many years of studying the mechanism and regulation of BC biosynthesis in the bacterium *Acetobacter xylinum* (*K. xylinus*) [84,85]. The CS complex is encoded by cellulose synthase operons known as bcs operons, which regulate intracellular biosynthesis, extracellular transport across the cellular membranes, and the in vitro assembly of cellulose fibrils into highly ordered structures [37]. In the past few decades, extensive work has been carried out to characterize BC biosynthesis [83,84,85,86,87,88]. Xanthan is produced using a Wzy-dependent pathway which has several steps, including the synthesis of exopolysaccharide precursors, repeat-unit assembly on a lipidcarrier located at the cytoplasmic membrane, membrane translocation to the periplasmic face, polymerization by a block-transfer mechanism involving Wzy polymerase, and export (Figure 1D) [36].

## 3. Strategies for Cost-Effective Production of Xanthan and BC

Like other EPSs, xanthan and BC yield and quality depend on many parameters and variables, including culture medium composition, temperature, pH, oxygen transfers, agitation rate, inoculum volume, cultivation method, fermentation duration, etc. Statistical methods have been applied systematically to optimize xanthan and BC production parameters [89,90,91,92,93,94,95,96,97,98]. A recent interesting review by Rocha et al. (2023) presents the main BC biosynthesis processes and strategies to optimize its production at industrial scale for the purposes of bioeconomy [99]. Xanthan is produced through submerged aerobic fermentation under dynamic conditions. The optimum temperature and pH for xanthan production is 28–30 °C and pH 7–8, respectively. Various types of bioreactors have been used for xanthan production, but mechanically stirred bioreactors are the most common. In stirred reactors, the rate of oxygen mass transfer is affected by the air flow rate and stirrer speed. The air flow rate is usually maintained at a constant level, typically 1 L/L min. The stirrer speed usually varies during culture from lower values (200–300 rpm) at the beginning of the fermentation to higher values (400–600 rpm) later on. The fermentation is to be followed by the thermal inactivation of the bacterium *X. campestris*, since the bacterium is phytopathogenic. The next step is xanthan precipitation, which proceeds using a solvent, like ethanol, methanol, isopropyl alcohol, acetone, etc. Precipitated xanthan should be dried and ground to obtain a powder. Figure 2A shows xanthan production in agitated conditions. BC is also produced by aerobic fermentation, although, unlike xanthan, under both dynamic and static conditions (Figure 2B,C). The method selection depends on the final BC applications as well as the required morphological, mechanical, and physicochemical characteristics [100]. The static cultivation method is a traditional approach for BC production, and it has become widespread. According to the method, bacteria are grown in the containers with a growth medium, usually for 7–14 days at 28–30 °C and a pH of 4–7. A BC gel film forms on the medium surface, its size depending on the surface area of the medium (Figure 2(Ba)). After purification with sodium hydroxide by heating and further washing deionized water until the pH became neutral, the film becomes colorless and transparent (Figure 2(Bb)). Figure 2(Bc) shows BC micromorphology, which exhibits a nanoporous three-dimensional network structure with a random arrangement of ribbon-shaped fibrils. The high cost and low rate of production are the two main problems in static culture systems. However, BC produced using static conditions has more advantages in biomedical fields due to having a higher crystallinity, a linear structure, good flexibility, and elasticity. Agitated cultivation is the preferred method for BC industrial production. Less space is required in agitated cultivation compared to the static condition, and the aeration rate is higher during BC production. Different types of bioreactors are utilized for this purpose, including stirred tank bioreactors, airlift bioreactors, rotating disk reactors, rotary biofilm contactor reactors, and reactors with silicone membranes [100]. BC obtained under dynamic conditions has a high water-holding capacity but usually a lower crystallinity. Figure 2C shows BC produced by *K. sucrofermentans* B-11267 in dynamic conditions using HS medium (a), whey (b), and thin stillage (TS) (c), and an AFM image of the cellulose microfibrils is shown.

### 3.1. BC and Xanthan Production from Wastes

Recently, many studies have focused on cheap nutrient sources, diverse strains of producing microorganisms, as well as on improving their culture conditions for the purpose of cost-effective xanthan and BC production [45,100,101,102,103,104,105]. The high cost of fermentation media is the limiting factor for the economic production of BC and xanthan. Nearly 30% of the process cost is likely to be the cost of a nutrient medium [106]. Among culture media, the most frequently used one is a chemically defined media. For example, the standard medium for obtaining BC is the Hestrin–Schramm (HS) medium, which contains expensive components, e.g., glucose, peptone, yeast extract, citric acid, and disodium phosphate. Other carbon sources such as fructose, maltose, xylose, sucrose, galactose, and others can be an alternative to glucose in the HS medium to optimize and increase BC production [107,108]. However, they are not economically viable for industrial scale production either [98]. Many researchers point out that the most promising medium for producing xanthan is sucrose supplemented with salts [92,109]. Also, starches (corn starch, potato starch), which have a relatively high cost, can serve as a source of carbon for xanthan production [110].

Whereas cost is a limiting factor in EPS production, many investigations are intent on using industrial wastes and by-products as a cost-effective substrate to produce BC and xanthan [45,101,102,103,104,105]. Kadier et al. (2021) divided industrial wastes into six groups: (1) brewery and beverages industries wastes; (2) agro-industrial wastes; (3) lignocellulosic biorefineries, pulp mills, and sugar industries wastes; (4) textile mills; (5) micro-algae industry wastes; and (6) biodiesel industry wastes [101]. Currently, several reviews on BC and xanthan production from food and agro-industrial wastes have been reported [45,101,102,103,104,105]. So, a recent review by Khan et al. (2023) has discussed the possibility of producing BC from different foods and agro-industrial wastes, describing various fermentation methods used for BC production as well as the biochemical and molecular regulation of BC production during its microbial synthesis [102]. Figure 3 shows a schematic overview of BC and xanthan production from different food and agro-industrial wastes.

The main results on the use of some food and agro-industrial wastes for BC production by *Komagataeibacter* sp. are summarized in Table 1. Table 2 summarizes some of the food and agro-industrial wastes that serve as a potential source of xanthan production.

After the industrial processing of fruits in juice factories, a huge amount of fruit peels remain, such as the peels of pomegranate, mango, pineapple, banana, citrus, and other fruits [162]. They are inexpensive raw materials for obtaining bacterial EPS. For example, Hasanin et al. (2023) reported about the sustainable BC production by *Achromobacter* using mango peel waste [75]. In addition, there were reports on BC production using pineapple and watermelon peels [163], citrus peel and pomace [134,164,165], and banana peel [166]. Abdelraof et al. (2019) reported about the production of BC from potato peel waste [162]. Xanthan was also produced from fruit and vegetable peels. So, Mohsin et al. (2018) optimized xanthan biosynthesis in a 15-L fermenter to achieve maximum polysaccharide production from 30.19 g/L of orange peels [146]. Shiram et al. (2021) reported a xanthan yield of 40.88 g/L and 31.4 g/L using food wastes and carrot and pumpkin peels, respectively [104]. This year, Chaiyachet et al. (2023) reported on BC production from *K. xylinus* TISTR 1011 and *K. nataicola* TISTR 975 using yam bean juice as a nutrient source [139]. Gorgieva et al. (2023) first reported on utilizing a grape pomace hydrolysate without enzymatic treatment as a sole culture medium for efficient BC production with the recently described species *K. melomenusus* AV436T [117]. The membranes were synthesized in only 4 days of bacteria culturing with a BC yield of 1.24 g/L.

Many studies have focused on using crude glycerol, which is a by-product of the biodiesel industry, to produce valuable products. It will both reduce the cost of xanthan and BC and resolve the excess glycerol problem. However, crude glycerol contains various impurities (namely: methanol, ethanol, inorganic salts, metals, long chain fatty acids, and soaps) to which producing microorganisms must be tolerant [155]. The article by Rončević et al. (2020) confirmed the possibility of xanthan production on a crude glycerol-based medium as a sole carbon source using different *X. campestris* strains [155]. The results obtained indicate that all the strains considered can be used as xanthan-producing microorganisms on a crude glycerol-based medium, and the highest xanthan concentration (7.67 g/L) was obtained using the Xp 3–1 strain. Wang et al. (2017) produced xanthan by a mutant strain *X. campestris* CCTCC M2015714 with glycerol as the sole carbon source [167]. The strain that could use glycerol to produce high-transparency and low-viscosity xanthan was obtained by adaptive evolution, and the yield of xanthan reached 11.0 g/L [154]. BC is the only commercially important exopolysaccharide whose synthesis on glycerol has been performed for a relatively long time, and therefore there is an abundance of accumulated data [168]. The review by Zikmanis et al. (2021) summarized the current knowledge on using glycerol to obtain BC, including information about producer cultures, composition of culture media, cultivation conditions, and productivity of bioprocesses [168]. Glycerin easily penetrates into producers’ cells by facilitated diffusion, while other renewable and cost-efficient feedstocks, such as molasses or cheese whey, require an enzymatic or chemical hydrolysis of relevant disaccharides (sucrose and lactose, respectively). The efficiency of using glycerol is also determined by the fact that the necessary enzymatic stages of its catabolism are well represented in the metabolic network of BC producers [169]. Although glycerol consumption is lower than that of glucose, it better contributes to BC production, since 48% of glycerol is used for the biosynthesis of this polymer, whereas glucose contribution to this process reaches 40% [170]. This factor is likely explained by the non-accumulation of organic acids when glycerol is used as a carbon and energy source for microorganism growth. Glycerol can be used in a purified and in a crude, unpurified form. Purified glycerol is predominantly a part of the composition of nutrient media, although a sufficiently high BC yield is also achieved when using crude glycerol [145,169,171,172]. So, Dikshit and Kim (2020) reported the highest BC production (3.40 g/L) to be observed at 50 g/L of initial pure glycerol concentration and 2.93 g/L with the same amount of crude glycerol [145].

Dairy, sugar, and alcohol industrial wastes such as whey, molasses, and stillage were also studied and considered as alternative substrates to enhance EPS production. Recently, several reviews on the production of biodegradable microbial polymers from whey have been reported [173,174,175]. This byproduct of the dairy industry has a high biochemical oxygen demand (BOD) that causes various environmental problems. On the other hand, it contains about 55% of the nutrients from the original milk, including carbohydrates such as lactose (45–50 g/L) and small amounts of galactose, glucose, arabinose, and lactulose, as well as soluble proteins (6–8 g/L), lipids (4–5 g/L), mineral salts (8–10% of dried extract), amino acids, vitamins, and organic acids such as lactic acid and citric acid [176,177,178]. Some researchers have studied the effect of whey on BC production. For example, Carreira et al. (2011) and Tsouko et al. (2015) observed a low BC yield from cheese whey [179,180]. However, Suwanposri et al. (2014) obtained BC in the amount of 4.10 g/L on day 7 of the static cultivation of *Komagataeibacter* sp. PAP1 using soya bean whey [98]. In addition, Revin et al. (2018) obtained BC in the amount of 5.45 g/L on the third day of cultivation of *K. sucrofermentans* B-11267 under agitated conditions using cheese whey without any pretreatment or adding other nitrogen sources [112]. The crystallinity of BC was 50.2%. Recently, Salari et al. (2019) and Brugnoli et al. (2023) have investigated the cheese whey treated with β-galactosidase as potential feedstock for producing BC [114,175]. So, Salari et al. obtained BC in the amount of 3.55 g/L on day 14 of the static cultivation of *G. xylinus* PTCC 1734 [114]. Brugnoli et al. observed the highest BC yield of 0.3 g BC/g of a carbon source consumed culturing *Komagataeibacter* sp. K2G39 isolated from green tea kombucha using cheese whey with an initial lactose concentration of 40.12 g/L, which was almost completely hydrolyzed after being treated with β-galactosidase [175]. The crystallinity of BC was 53.55%. Most of *Xanthomonas* wild species do not consume lactose because of their low β-galactosidase activity. To solve the problem, several efforts have been made to transfer the β-galactosidase genes by plasmid or phage vectors to the *Xanthomonas* genome and force the bacteria into lactose consumption [181]. Mutagenesis methods have also been used to develop mutated strains for lactose consumption [182,183]. But genetic modification is usually associated with some disadvantages, including the production of antibiotic-resistant strains and strain instability [182]. The risk of genetically modified organisms in food industries may also be a suppressing factor. To solve the problem, the best option is to use the native strains which have the natural ability to use lactose for xanthan production. So, Ramezani et al. showed the highest amount of produced xanthan by the *X. citri* NIGEB-K37 strain in a lactose-based medium (lactose 10%) to be 14.26 g/L [71]. Moravej et al. (2020) increased the amount of xanthan to 18.4 g/L in the cheese whey medium without adding more lactose using the *X. citri* NIGEB-K37 [157].

Molasses is one of the most studied waste products for EPS production [113,116,159,184,185,186,187,188]. It is a by-product of the final stage of crystallization in sugar production and contains about 50% total sugars by dry weight. Sucrose cannot be transported across the cell membrane and is hydrolyzed in a periplasm to glucose and fructose by α-glucosidase, which may have different activities depending on the producer. Therefore, such substrates are usually subjected to hydrolytic treatment using enzymes or less expensive chemical hydrolysis with mineral acids. Bae and Shoda [184] and Çakar et al. [185] reported that BC production can be increased drastically by H_2_SO_4_-heat treatment of sugar beet molasses. Also, Abol-Fotouh et al. (2020) suggested the preliminary thermal acid treatment of molasses to break down the contained sucrose to glucose and fructose [186]. Bae and Shoda found that the highest production of BC was obtained at 20 g/L of total sugar concentration of sugar beet molasses [184], while the medium containing 5% (*w*/*v*) total sugar in sugar beet molasses led to the highest BC production in the study by Jung et al. [189]. Salari et al. (2019) showed that sugar beet molasses without any supplementation can be used as a single cheap carbon source for BC production by *G. xylinus*. The maximum BC production was 4.56 g/L in sugar beet molasses with an initial sugar concentration of 20 g/L after 14-day fermentation under static conditions, which was higher than that on the standard HS medium (3.26 g/L) [114]. Revin et al. (2021) reported that BC production by *K. sucrofermentans* H-110 was 2.9 g/L in sugar beet molasses with an initial sugar concentration of 25 g/L after 5 days of cultivation under static conditions, which is almost two times as high compared to the standard HS medium (1.6 g/L) [113]. The crystallinity of BC formed on the molasses medium was 83.02%, which was higher than on the HS medium. Moreover, Revin et al. (2021) obtained the highly efficient strain *X. campestris* M 28, which produced up to 28 g/L of xanthan on a molasses medium [159]. Öz and Kalender developed a new static cultivation system called a series static culture to eliminate the air limitation problems encountered in a conventional static culture [190]. Fermentation experiments were carried out using *G. xylinus* NRRL B-759 and sugar beet molasses at 30 °C and an initial pH of 5.

Stillage is the byproduct resulting from ethanol production. For every liter of ethanol produced, molasses-based distilleries usually generate about 8–15 L of stillage characterized by high chemical oxygen demand (COD) and biochemical oxygen demand (BOD) [191]. The thin stillage (TS) is the aqueous byproduct generated from the distillation of ethanol following the fermentation of starch or sugar crops during the ethanol production process. The fermentation and distillation processes of the feedstocks generate the whole stillage, which contains solids from the grain along with added yeast. The whole stillage is then centrifuged to separate the liquid component, called thin stillage, and the solid component, called wet distillers’ grain [192]. TS contains organic and inorganic compounds, some of which may be valuable products for EPS production. Ratanapariyanuch et al. studied the composition of the wheat thin stillage by HPLC and demonstrated it to contain dextrin (8.47–11.65 g/L), glycerol (2.39–7.87 g/L), lactic acid (5.07–7.41 g/L), acetic acid (0.56–2.72 g/L), succinic acid (0.63–0.93 g/L), ethanol (0.23–1.31 g/L), maltotriose (0.14–1.10 g/L), maltose monohydrate (0.03–1.05 g/L), glycerophosphorylcholine (0.91–1.11 g/L), and betaine (0.8–1.03 g/L) [193]. The availability of TS to increase the BC yield was studied [112,194,195]. Organic acids, glycerol, and ethanol are known from the literature to have a positive effect on BC production [99,112]. For instance, rice wine stillage containing organic acids was applied as an additive to the HS medium to increase cellulose yield. The largest BC amount (6.31 g/L) was obtained in the HS medium diluted with 50% stillage [194]. Revin et al. used wheat TS without any pretreatment or adding other nitrogen sources to reduce BC cost [112]. The studies showed that the maximum BC yield (6.19 g/L) was observed in the TS for 3 cultivation days under agitated conditions, and that was nearly three times as high compared to the BC yield in the HS medium (2.14 g/L).

To reduce the cost of BC production, a number of researchers also assessed the availability of using wastewater from acetone–butanol–ethanol (ABE) fermentation. ABE fermentation wastewater contains fermentable sugars, organic acids, and alcohol compounds. Typically, organic acids such as acetic acid and butyric acid are the main by-products of ABE fermentation, while the presence of alcohol compounds is associated with the incomplete distillation of the ABE fermentation broth. Xiong et al. (2015) analyzed ABE fermentation wastewater by HPLC and demonstrated that xylose and glucose were the two main residual sugars in it, their concentrations being 0.61 and 0.26 g/L, respectively [196]. As for organic acids, acetic acid and butyric acid were the two main kinds of them, and their concentrations were 1.70 and 1.00 g/L, respectively. Furthermore, some alcohol compounds (g/L, ethanol 1.00, and butanol 1.15) were also present in the wastewater. The total nitrogen concentration in the wastewater was extremely low (merely about 48.3 mg/L). Overall, the ABE fermentation wastewater had a high COD value (18,050 mg/L). The previous literature reports demonstrated the secondary substrates or supplements (organic acids, ethanol, butanol) to be essential to facilitate BC production [99,112,196,197]. Huang et al. (2015) used the wastewater generated by fermentation broth distillation after ABE fermentation without any pretreatment or adding nutrients as the substrate for BC production using *G. xylinus* [197]. After 7-day fermentation in static culture, the highest BC yield (1.34 g/L) was obtained. The carbon sources, including sugars (glucose and xylose), organic acids (acetic acid and butyric acid), and alcohol compounds (ethanol and butanol), were utilized by *G. xylinus* simultaneously during fermentation.

Nowadays, lignocellulosic wastes are in focus as renewable and abundant substrates to produce various EPSs [198]. However, microorganisms cannot utilize them directly as a carbon source; therefore, pretreatment and hydrolysis of lignocellulosic materials are necessary [199]. The cellulolytic and hemicellulosic fractions used for EPS production are promising due to their high carbon content [153]. Corncob is a low-cost substrate with a high potential to provide fermentable sugars (glucan, xylan, arabinan) [152,200]. Different fractions (cellulose, hemicellulose, and lignin) extracted from corncob can be alkaline. The hemicellulosic fraction extracted by alkali is cheaper compared to corn starch. Jesus et al. (2023) evaluated the potential of the hemicellulosic fractions obtained by the alkaline hydrolysis of corncob and used as a carbon source, macro, and micro-nutrients in xanthan production, using different strains of *X. campestris* (629, 1078, 254, and S6) [152]. The findings indicate that strain 629 provides the higher yield (8.37 g/L) when using a fermentation medium containing saccharose (1.25%), hemicellulose fractions (3.75%), and salts. Soleimanpour et al. (2018) proposed a broomcorn stem hydrolyzed by sulphuric acid as a low-cost and widely available carbon source for xanthan production. The maximum yield of the polysaccharide was 8.9 g/L [91]. There were several studies reported on the feasibility of using different wastes with a lignocellulosic content in BC production, including corncob and sugarcane bagasse [119], oat hull-derived enzymatic hydrolyzates [201,202], enzymatic hydrolysate of wheat straw [123], pulp mills, and lignocellulosic wastes [203,204]. Bagasse is a fibrous biomass generated from sugarcane processing, and it is the material that remains after extracting juice from the sugarcane stalks. It is composed of cellulose, hemicellulose, and lignin, making it a good candidate for EPS production. Microcrystalline cellulose is present in different agro-industrial wastes such as walnut shells, corncob, and sugarcane bagasse [205]. Moreover, to make xanthan production cost-effective, agricultural and food wastes such as tapioca pulp [147], waste bread [149], kitchen waste [150], jackfruit seed powder [151], cocoa husk [153], fermenting shrimp shell [161], sugarcane bagasse [206], rice bran [207], chicken feathers [208], coconut shell, potato crop [209], winery wastewater [210], and demerara sugar [92] were used.

### 3.2. Technologies for Cost-Effective Production of Xanthan and BC

#### 3.2.1. Co-Cultivation of EPS-producing Bacteria

Another strategy for efficient EPS production can be the co-cultivation of EPS-producing bacteria. A number of publications point out the positive effect of some water-soluble polysaccharides on BC yield [211,212]. For example, Seto et al. (2006) first reported that the co-cultivation of the two bacteria *G. xylinus* and *Lactobacillus mali* in corn steep liquor/sucrose liquid medium resulted in a threefold higher cellulose yield when compared to monoculture [211]. Liu et al. (2019) examined a novel fermentation process which consists of co-culturing *G. hansenii* ATCC 23769 with *Escherichia coli* ATCC 700728 under static conditions and producing BC pellicles with enhanced mechanical properties [213]. The authors suggested the mannose-rich EPS synthesized by *E. coli* to be incorporated into the BC network and affect the aggregation of co-crystallized microfibrils. The BC pellicles exhibited a Young’s modulus of 4874 ± 1144 MPa and stress at a break of 80.7 ± 21.1 MPa. Nazarova et al. (2022) reported that the co-cultivation of the bacterial cellulose producer strain *K. sucrofermentans* B-11267 and the dextran producer strain *L. mesenteroides* VKM B-2317D doubled the yield of BC compared to monoculturing from 2.64 g/L to 5.99 g/L, respectively [214]. The increase in BC yield is likely due to the fact that the dextransucharase, which is formed by bacteria of the genus *Leuconostoc*, enables the quick breaking down of the sugars contained in molasses. Dextran formed by bacteria can also contribute to BC formation. The increase in product yield might be associated with fructose, which is formed when sucrose is broken down by the enzyme. Fructose is known to be a good source of carbon to cultivate BC producers [14]. According to the literature data, as a result of fructose metabolism, fewer organic acids are formed and there is no strong acidification of the environment as when using glucose, which is converted into gluconic acid. The co-cultivation of EPS-producing bacteria can also be considered as a method for obtaining biocomposite materials. Including additives in the culture media during BC biosynthesis is a traditional method to produce BC-based composites. Brugnoli et al. (2023) developed a co-culture system combining BC producers of the genus *Komagataeibacter* and hyaluronic acid producers of the *Lactocaseibacillus* genus and highlighted a higher BC yield and the incorporation of hyaluronic acid into the composite [215]. The presence of hyaluronic acid improved the water-holding capacity of the composites, resulting in a decrease of BC crystallinity.

#### 3.2.2. Biocatalytic Technologies

Biocatalytic technologies seem to be promising to obtain bacterial EPS. Efremenko et al. (2022) summarized in their review the information on the currently known biocatalytic synthesis of microbial polysaccharides and discussed the prospective research development in the field of biocatalysis [216]. The degree of carbohydrate substrate conversion into a biopolymer can grow by improving the specific activity of enzymes involved in the synthesis and regulating the pathways in EPS precursor biosynthesis. Immobilized cells can significantly increase the productivity and stability of biocatalysts [216]. An important advantage of cell immobilization is the capability of their long-term functioning, which enables a significant increase in the overall efficiency of EPS production. In addition, immobilized cells, when in the QS state, can withstand high concentrations of toxic substances compared to free cells. QS activates EPS synthesis [216,217,218,219,220] as protective and stabilizing and reserves substances for highly concentrated microbial populations, as it is a natural mechanism to increase the amount of the biopolymers and can be used as a nature-like technology in their industrial production. Several studies have been carried out on the use of immobilized cells for EPS biosynthesis in which the high efficiency of the approach has been proven. Examples of immobilization of EPS-producing cells among lactic acid bacteria of the genus *Lactobacillus* are known [221,222]. The productivity of such cells exceeded the productivity of free cells. The immobilization of bacteria of the genus *Xanthomonas* on granules based on calcium alginate also showed a higher xanthan yield compared to free cells [158]. The capability of cell immobilization of BC producers *A. xylinum* in Ca-alginate gel [223] and *K. xylinum* B-12429 in cryogel based on polyvinyl alcohol (PVA) [224] was demonstrated. *K. xylinum* B-12429 cells immobilized in PVA cryogel synthesized BC 1.6 times more than in the suspension culture. At the same time, the BC films had a higher tensile strength, a 30% greater thickness, and a higher polymerization degree. Rahman et al. (2021) reported for the first time on BC production using a natural loofah sponge as a scaffold for *G. kombuchae* immobilization [225]. The fermentation was carried out using free cells and immobilized cells under shaking and static cultivation for 15 days. The maximum BC concentration of 15.5 ± 1.65 g/L was obtained in a medium containing immobilized cells with shaking.

Cell-free systems for EPS biosynthesis also show great development prospects [226,227,228]. The cell-free systems may be a possible solution to the limitations faced by traditional EPS production processes, such as low yield and productivity, the production of byproducts and secondary metabolites, and high downstream processing costs. In addition, cell-free systems expand the possibilities of obtaining biocomposite materials in situ, for example, when obtaining materials with antibacterial properties, since antibiotics and other substances with antibacterial action will inhibit the growth of bacterial cells. A recent review by Ullah et al. (2023) presents a comprehensive overview of the development of cell-free systems, ranging from crude cellular extracts of various organisms to advanced cell-free designs, based on the principles of synthetic biology and using genetic and metabolic engineering approaches [228]. The review provides information on developing a cost-effective cell-free system, including the cost of cofactors, enzymes, raw materials, process efficiency and scalability, and potential directions for its large-scale implementation in the future.

#### 3.2.3. Genetic and Metabolic Engineering

Another strategy for the cost-effective production of xanthan and BC is to isolate new bacterial strains from natural sources and obtain highly productive strains by genetic and metabolic engineering. With the help of genetic engineering, new strains with modified or introduced enzymatic activity can be developed, which can expand the range of inexpensive substrates available for production, increasing their degree of transformation into EPSs. A growing number of studies focused on investigating the mechanisms involved in BC and xanthan biosynthesis, metabolic modeling, and genetic analysis have been applied to enable improving its production on a large scale [35,37,99,154,229,230,231,232,233,234,235]. Currently, several reviews on the genetic modification of BC-producing bacteria have been reported [84,85,86,229,230,231,232]. Singhania et al. (2021) reported various mechanisms for genetic modifications to achieve the desired changes in BC production as well as its characteristics [230]. The authors conveyed the lack of studies on a genetic modification for BC production to be due to the limited information on the complete genome and genetic toolkits; however, over the past few years, the number of studies in this area has increased, since the whole genome sequencing of several bacterial strains has been obtained. Genetic modification can improve BC production either by blocking genes responsible for the synthesis of metabolic by-products or by overexpressing the genes involved in polysaccharide biosynthesis. In addition to the above-mentioned advantages, there are some challenges, including methodological problems of transformation and problems concerning the complexity of the regulatory process when each gene can express a protein having more than one function [230]. However, there have been several efforts made in genetic engineering aimed at BC-producing bacteria. For example, Kuo et al. created a *G. xylinus* mutant by knocking out the membrane-bound glucose dehydrogenase gene, which led to BC synthesis from glucose without generating gluconic acid and a 40% increase in polysaccharide production [232]. A new, stable, and efficient plasmid-based expression system of recombinant BC in the *E. coli* DH5_ platform has recently been developed [233]. The review by Buldum et al. (2021) presented the potential of ‘modern genetic engineering tools’ and ‘model-driven approaches’ on improving the yield of BC, altering the properties, and adding new functionality [86]. Until recently, efficient techniques for the generation of markerless modifications in the genome of BC-producing strains were not available. Most genetic studies were conducted by transposon mutagenesis, which can limit the interpretability of the results due to polar effects and other artefacts [86]. The markerless deletion system for a high-yield cellulose-producing bacteria has advantages over transposon mutagenesis as it avoids possible polar effects and allows better biotechnological tuning of BC production in the future [87]. Recently, Bimmer et al. (2023) reported the construction of various mutants, their phenotypical, transcriptomic, and proteomic characterization, as well as the quantification and analysis of the synthesized BC by scanning electron microscopy and physicochemical parameters [87]. Furthermore, Yang et al. (2023) constructed a recombinant strain of *K. xylinus* ATCC 23770 for the production of BC from mannose-rich resources [234]. This strategy aimed at the modification of the mannose catabolic pathway via the genetic engineering of bacterium through the expression of mannose kinase and phosphomannose isomerase genes from the *E. coli* K-12 strain. The comparison showed that with mannose as the sole carbon source, the BC yield from the recombinant strain increased by 84%, and its tensile strength and elongation were increased 1.7 fold, while Young’s modulus was increased 1.3 fold. Jang et al. demonstrated that the *K. xylinum* strain overexpressing the *E. coli* glucose 6-phosphate isomerase gene produced 3.15 g/L of BC, which was 115.8% higher as compared to the 1.46 g/L obtained from the control strain [235]. Using genetic engineering for EPS production is a promising alternative to improve production on an industrial scale. However, it should be noted that despite numerous investigations to develop productive strains, none of them has so far resulted in mutant strains which could comply with the requirements of large-scale biotechnological production [216]. In addition, using genetically modified microorganisms on an industrial scale generally has a number of considerable limitations—chiefly, environment-associated limitations.

## 4. Applications of BC and Xanthan

Despite a number of differences in their structure and properties, xanthan and BC have found wide application in similar fields of medicine, technology, and industry (Figure 4). Their promotion is due to their unique beneficial properties such as biocompatibility, biodegradability, non-toxicity, a high degree of polymerization, water retention, and the ability for gelation. Thus, they can be commercially applied in food, pharmaceutical, cosmetic, chemical, textile, oil, and gas industries as thickeners, emulsifiers and suspension stabilizers, flocculants, and additives to improve the quality of different products. The biocompatibility and functional characteristics of BC and xanthan are key factors promoting their application in biomedicine, e.g., tissue engineering, wound dressing, and drug delivery systems. Recently, many reviews describing the use of BC [7,37,52,53,54,55,56,57,236,237,238,239,240,241,242] and xanthan [48,63,89,243,244,245] in various fields have been published.

Xanthan, which was discovered in the 1950s, belongs to one of the earliest marketed bacterial exopolysaccharides certified for food use in the USA [246]. This polymer is environmentally friendly and non-toxic and therefore is used in the food industry as a thickener, stabilizer, and suspending agent in many foods and in the structure of biodegradable food packaging [48,79,243,244,245,246]. BC is also used as an additional thickener, a suspending or stabilizing agent in foods, or directly in food as an ingredient in fiber-enriched low-calorie and low-cholesterol diets, as well as the material for food packaging [247,248,249,250,251,252]. In 1992, the Food and Drug Administration (FDA) approved BC to be safe, and in 2019, the species *K. sucrofermentans* was included in the list of Qualified Presumption of Safety (QPS) recommended biological agents and intentionally added to food [249].

BC and xanthan are promising materials for biomedical applications since they are biocompatible polymers and not cytotoxic. BC-based materials for medicine have been produced for a relatively long time. Back in the early 1980s, the American pharmaceutical company Johnson & Johnson proposed using BC films to treat superficial wounds. Recently, several commercial medical BC-based materials have been obtained: Biofill^®^ (Curitiba, Brazil) and Bioprocess^®^ (Curitiba, Brazil) for burns and ulcer therapy, Gengiflex^®^ (Curitiba, Brazil) to treat periodontal diseases, Dermafill^®^ (Londrina, Brazil) for effective wound and ulcer healing, Membracel^®^ (Curitiba, Brazil) to treat venous leg ulcers and lacerations, xCell^®^ (New York, NY, USA) for chronic wounds therapy, EpiProtect^®^ (Royal Wootton Bassett, UK) for burn wounds, and Nanoskin^®^ incorporated with silver ions (São Carlos, Brazil) [236,253]. BC and xanthan have great potential to be used in medicine as a biomaterial for wound dressing [254,255,256,257,258,259,260,261,262,263,264,265,266], drug delivery systems [267,268,269,270,271,272,273,274,275,276,277,278,279,280,281,282,283,284,285,286,287,288,289], and tissue engineering [290,291,292,293,294,295,296,297,298,299,300,301,302,303,304,305,306,307,308,309,310,311,312,313,314,315,316,317,318,319,320,321,322,323,324,325,326,327,328]. Recently, many reviews on BC-based materials for biomedical applications have been reported. This year, reviews by Qian et al. (2023) and Jadczak et al. (2023) summarized the state-of-the-art application of functional BC-based materials in biomedical fields [52,241]. Also, the recent review by Tang et al. (2022) discussed some of the biomedical applications that use BC, including wound healing, drug delivery, tissue engineering, and tumor cell and cancer therapy [242]. The publications on using xanthan to obtain medical materials have appeared relatively recently. So, in the last two decades, researchers have taken an interest in its future use in drug delivery, tissue engineering, as well as biocomposites with regenerative and antibacterial properties [261,262,263,264,265,266,274,275,276,277,278,284,285,286,287,288,289,321,322,323,324,325,326,327,328].

Wounds need an appropriate wound dressing to help prevent bacterial infection and accelerate wound closure. Wound dressing materials fabricated using biocompatible polymers have become quite relevant in medical applications [267]. BC is a biopolymer that is commonly used for wound dressings due to its high biocompatibility, good flexibility, strong water-holding capacity, vapor permeability, elasticity, and non-toxicity [254,255]. Recently, a review by Horue et al. (2023) provided information on BC-based materials as dressings for wound healing [53]. The authors reported the main characteristics of different BC structures such as films, membranes, fibers, etc., as well as recent advances in BC-based composites. Furthermore, the review by de Amorim et al. (2022) offers a summary of advances in the use of BC in composites and polymeric blends for drug delivery systems and wound healing [258]. The review by Meng et al. (2023) introduces recent advances in BC-based antibacterial composites for the treatment of wound infection, including classification and preparation methods of composites, the mechanism of wound treatment, and commercial applications [259]. Pasaribu et al. (2023) developed bioactive BC-based wound dressings for burns by impregnating collagen via an in situ method followed by immersing chitosan via an ex situ method into BC fibers [260]. In vivo tests indicated that BC/collagen/chitosan wound dressing supported the wound healing process for second degree burns. Tang et al. (2022) developed hydrogel wound dressings using xanthan gum and polyacrylamide [261]. With the combination of the polyacrylamide network and the xanthan network, the composite hydrogels showed high tensile strength, stretchability, excellent water uptake efficiency, outstanding biocompatibility, universal adhesion, and self-healing ability [261]. Singh et al. (2022) developed polyvinyl alcohol copolymerized with xanthan gum/hypromellose/sodium carboxymethyl cellulose dermal dressings functionalized with biogenic nanostructured materials for antibacterial and wound healing applications [262]. Recently, Gutierrez-Reyes et al. (2023) investigated novel hydrogels of semi-interpenetrating polymeric networks based on collagen and xanthan gum for wound healing applications [263]. The increment of xanthan in the hydrogel (up to 20 wt.%) allows for improvement in the storage module, resistance to thermal degradation, and the slowing of the rate of hydrolytic and proteolytic degradation, allowing the encapsulation and controlled release of molecules such as ketorolac and methylene blue. Recently, Unalan et al. (2023) developed three-dimensional (3D)-printed sodium alginate–xanthan gum hydrogels containing phytotherapeutic agents with antioxidant and antibacterial activity as multifunctional wound dressings [264]. Liang et al. (2023) prepared 3D-printed antibacterial hydrogels with benzyl isothiocyanate using xanthan gum, locust bean gum, konjac glucomannan, and carrageenan for burn wound healing [265]. Alves et al. produced a thermo-reversible hydrogel composed of xanthan–konjac glucomannan (Figure 5B) [266]. The authors demonstrated the potential of composite hydrogels to improve the wound healing process by promoting fibroblast migration, adhesion, and proliferation [266].

Drug delivery systems are used for the targeted delivery and/or controlled release of therapeutic drugs and have the advantage of reducing side effects, improving therapeutic effects, and possibly reducing drug doses [267,268]. Recently, EPSs have been considered as the ideal candidates for drug delivery systems due to their good biocompatibility, low immunogenicity, biodegradability, renewable sourcing, and easy modification [269]. In recent years, interesting reviews have been published characterizing the EPS-based materials used in drug delivery systems [269,270,271,272]. For example, the review by Qiu et al. (2022) introduced a variety of polysaccharide-based nanocarriers such as nanoparticles, nanoliposomes, nanomicelles, nanoemulsions, and nanohydrogels for diabetes treatment [270]. The review by Huo et al. (2022) summarized the latest research work on nanocellulose-based materials used in drug delivery [271]. The review by Lunardi et al. (2021) provides a comprehensive overview of the procedures for modifying and functionalizing nanocellulose to obtain carriers in drug delivery systems [272]. Chung et al. produced BC loaded with antibodies for optimizing checkpoint-blocking antibody delivery (Figure 5C) [273]. Recently, the review by Jadav et al. (2023) provided a comprehensive summary of current advances in xanthan modification to be used as an excipient in pharmaceutical formulation development, highlighting xanthan applicability to deliver various therapeutic agents such as drugs, genetic materials, proteins, and peptides [274]. The important characteristics of xanthan for drug delivery systems are high stability at a low pH, which helps protect a drug in gastric fluid from degradation, and the ability to control the drug release rate by changing the pH and ionic strength of the release medium. Different forms of xanthan, such as hydrogels, matrix tablets, films, microspheres, and mucoadhesive patches, are synthesized to deliver drugs in various diseases [274]. The review by Jadav et al. provides information on xanthan-based systems for the delivery of anti-diabetic drugs, anti-spasmodic drugs, immunosuppressive drugs, and drugs to treat inflammation, rheumatoid arthritis, gout, skin diseases, central nervous system-related disorders, obesity, glaucoma, and pulmonary diseases [274]. Xanthan-based materials are used to deliver antibacterial [275,276], antiviral [277], and antifungal [278] drugs. Moreover, BC has been used to deliver antibacterial and antiseptic agents [279]. BC and xanthan have been shown to be promising biomaterials for cancer treatment [280,281,282,283,284,285,286,287,288]. For example, Cacicedo et al. combined a BC hydrogel and nanostructured lipid carriers to use as an implant for the local drug delivery of doxorubicin in cancer therapy [281]. Zhang et al. developed BC-based composites with Fe_3_O_4_ magnetic doxorubicin-coated nanoparticles for breast cancer therapy [282]. Microspheres, hydrogels [284,285,286], pH-responsive nanoparticles [287], and nanogels [288] of xanthan were prepared for the delivery of anticancer drugs used to treat different cancers, including colon cancer. Recently, Anghel et al. (2023) developed novel xanthan-based materials as a delivery carrier for heparin [289].

Recently, the fabrication of xanthan and BC-based scaffolds, including composites and blends with nanomaterials, and other biocompatible polymers has received particular attention owing to their desirable properties for tissue engineering. BC has a huge potential in tissue engineering due to its favorable mechanical properties, biocompatibility, high hydrophilicity, crystallinity, purity, high degree of polymerization, and ultrafine porous fibrous collagen-like structure [290,291,292]. In the past few decades, many papers have been published on the use of BC in tissue engineering. Recently, the review by Raut et al. (2023) presented the latest modified/functionalized BC-based composites and blends as advanced materials in tissue engineering and summarized the latest updates on the production strategies and characterization of BC and its composites and blends [292]. BC-based composites have proven to be promising materials in cartilage [293,294,295,296,297,298,299,300], bone [301,302,303,304,305], soft tissue engineering such as blood vessels, adipose tissue, nerves, the liver, and skin [306,307,308,309,310,311,312,313,314,315,316,317,318,319,320]. The review by Jabbari et al. (2022) discussed the importance and essential role of BC-based biomaterials in neural tissue regeneration and the effects of electrical stimulation on cellular behaviors [312]. The review by Chen et al. (2022) summarized the application prospects of cellulose and its derivative-based hydrogels in biomedical tissue engineering [313]. A recent review by Fooladi et al. (2023) discussed the application of BC-based materials for cardiovascular tissue engineering [314]. Dydak et al. developed BC-coated Titanium-Aluminium-Niobium bone scaffold implants with low cytotoxicity against osteoblast and fibroblast cell cultures (Figure 5E) [320]. Zuliani et al. demonstrated that it is possible to differentiate stem cells from human amniotic fluid into chondrocytes when seeded directly in an efficient and low-cost chitosan-xanthan scaffold (Figure 5D) [321]. Bueno et al. obtained xanthan–hydroxyapatite hydrogel nanocomposites by precipitating hydroxyapatite in a xanthan aqueous solution. Nanocomposite hydrogels presented a porous structure and proved to be suitable for osteoblast growth [322]. Recently, Barbosa et al. (2023) produced chitosan–xanthan composite membranes, incorporating hydroxyapatite to be used in guided tissue and bone regeneration, in particular for periodontal tissue regeneration [323]. Souza et al. (2022) developed a chitosan–xanthan membrane associated with hydroxyapatite and different concentrations of graphene oxide for guided bone regeneration [324]. Furthermore, Souza et al. (2023) synthesized polymeric scaffolds of chitosan/xanthan/hydroxyapatite-graphene oxide nanocomposites associated with mesenchymal stem cells for regenerative dentistry applications [325]. Recently, Singh et al. (2023) fabricated biomaterial composed of xanthan and diethylene glycol dimethacrylate with impregnation of graphite nanopowder filler in their matrices for effective bone tissue regeneration purposes with improved biomineralization [326]. Piola et al. developed a crosslinked 3D-printable hydrogel based on biocompatible natural polymers, gelatin, and xanthan gum at different percentages to be used both as a scaffold for human keratinocyte and fibroblast cell growth and as a wound dressing (Figure 5F) [327]. In another study, Decarli et al. (2023) reported a reproducible bioprinting process followed by a successful post-bioprinting chondrogenic differentiation procedure using human mesenchymal stromal cell spheroids encapsulated in a xanthan gum–alginate hydrogel [328]. These results demonstrated a promising procedure to obtain 3D models for cartilage research and ultimately an in vitro proof-of-concept of their potential use as stable chondral tissue implants. Figure 5 shows a schematic overview of biomedical applications of xanthan and BC-based composites.
Figure 5Biomedical applications of xanthan and BC-based composites: BC gel film (**A**); xanthan-konjac glucomannan composite hydrogel for wound healing (**B**) (adapted from Ref. [266] (open access)); BC loaded with IgG for optimizing checkpoint-blocking antibody delivery (**C**) (adapted from Ref. [273] (open access)); chitosan–xanthan scaffold for chondrocytes growth (**D**) (adapted from Ref. [321] (open access)); BC-coated Titanium-Aluminium-Niobium bone scaffold (**E**) (adapted from Ref. [320] (open access)); 3D-printed gelatin–xanthan composite hydrogel for growth of human skin cells (**F**) (adapted from Ref. [327] (open access)); BC graft implanted in the porcine carotid artery (**G**) (adapted from Ref. [307] (open access)).
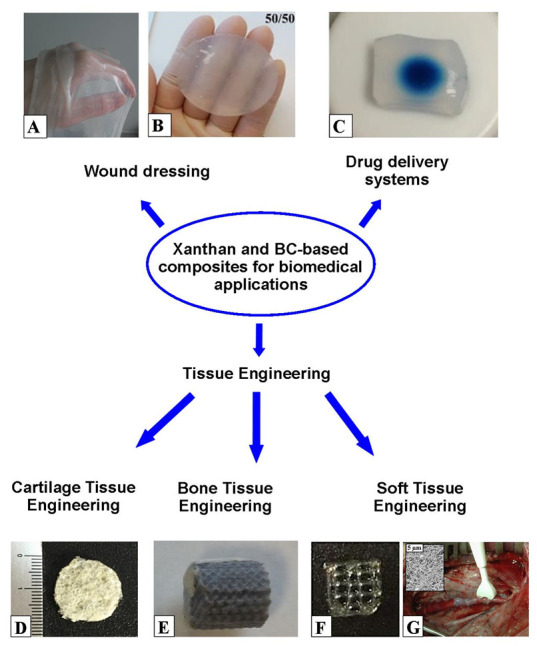



BC has many advantages when used as an adsorbent, including a large surface area, high mechanical properties, biodegradability, and high reactivity due to the presence of hydroxyl groups on the surface, which enables its chemical modification to interact with various pollutants, depending on its nature [46,329]. A number of BC-based adsorbents have been obtained for removing hazardous metals [330,331,332,333], fluorine [113], and organic pollutants, including dyes, pharmaceutical compounds, and petroleum products [334,335,336,337,338,339]. So, Salama et al. (2021) provided a comprehensive overview of the latest research results on nanocellulose-based materials for wastewater treatment, including adsorption, absorption, flocculation, photocatalytic decomposition, disinfection, etc., and discussed various approaches to their chemical modification [329]. Parizadeh et al. (2023) developed an effective colorimetric sensor that detects copper (Cu(II)) ions in solutions and solid states using anthocyanin extract from black eggplant peels embedded in BC nanofibers [340]. Xanthan functional groups are also able to bind heavy metals from aqueous solutions and effectively remove them. Xanthan can be used as a new green-based material to produce superabsorbents and remediate contaminated waters [36,243,245]. A recent review by Balíkova et al. has emphasized the prospects for using xanthan as an environmentally friendly adsorbent for water disinfection [36]. Sorze et al. (2023) have developed novel biodegradable hydrogel composites of xanthan and cellulose fibers that can be used both as soil conditioners and ground covers to stimulate plant growth and protect forests [341]. Recently, Guimarães et al. (2023) received a superabsorbent BC film produced from industrial residues of cashew apple juice processing [342]. Furthermore, bacterial EPSs have attracted interest for their applications, such as environmental bio-flocculants, because they are degradable and nontoxic. So, Sudirgo et al. (2023) showed xanthan to be a promising alternative as a coagulant aid for synthetic Congo red wastewater decolorization. The carboxylate group in xanthan could interact with polyaluminium chloride as the main coagulant, thus assisting the formation of larger flocs and resulting in improved coagulant performance [343].

Xanthan is widely used in enhanced oil recovery technology because of its high viscosity, pseudoplastic behavior, salinity stability, temperature, and alkaline conditions [344]. Furthermore, BC can be used for the microbial enhancement of oil recovery [77]. Furthermore, BC and xanthan can be used in nanoelectronics (sensors, optoelectronic devices, flexible display screens, energy storage devices, and acoustic membranes) [345,346,347,348,349,350,351,352,353,354,355,356]. Ionic conductive hydrogels have received widespread attention as ideal candidates for flexible electronic devices. Conductive polymers such as polypyrrole, polythiophene, and polyaniline and nanomaterials such as carbon-based nanomaterials, metal nanoparticles, or nanowires are used in the synthesis of conductive hydrogels. The recent review by Prilepskii et al. (2023) presented some recent developments in electrically conductive BC-based composites for applications in numerous areas, including electrically conductive scaffolds for tissue regeneration, implantable and wearable biointerfaces, flexible batteries, sensors, and EMI shielding composites [352]. The recent review by Pan et al. (2023) summarized the latest advances in BC hydrogel-based sensors, including strain, pH, electroactive, and thermal sensors [353]. Recently, Zhou et al. (2023) developed dual-network polyvinyl alcohol/polyacrylamide/xanthan gum ionic conductive hydrogels for flexible electronic devices [354]. Furthermore, Wu et al. (2022) prepared a novel ionogel with semi-interpenetrating poly (ionic liquids)/xanthan gum for highly sensitive pressure sensors [355]. Currently, Tomić et al. (2023) are developing self-healing and self-adhesive conductive nanocomposite hydrogels by multiple and diverse coordination connections between various polysaccharide-based modifiers (xanthan, arabic gum, sodium carboxymethyl cellulose), the poly(vinyl alcohol) network, and different graphene-based fillers [356]. Recently, xanthan and BC have received attention for their application in 3D printing technology [327,357,358,359,360,361,362,363,364,365]. Biopolymers as bioinks tend to be more profitable in terms of biocompatibility, nontoxicity, biodegradability, nonantigenicity, inertness, bio-adhesiveness, and adequate hemostasis compared to synthetic polymers. Xanthan has the required viscosity and shear thinning capacity, due to which it can function as a rheological modifier, thus improving 3D printing potential [357,358]. Recently, Li et al. (2023) developed a gelatin methacryloyl/alginate/polyethylene glycol dimethacrylate/xanthan gum hydrogel bioink system for extrusion bioprinting [361]. Xanthan improved the viscosity of the hydrogel system and allowed easy extrusion at room temperature and demonstrated solubility in ionic solutions such as cell culture medium, which is essential for biocompatibility. They have also developed an automated active mixing platform which allows for the high-quality preparation of hydrogel bioinks [362]. The use of BC and xanthan as bioinks for 3D printing has tremendous potential in tissue engineering and wound dressings [264,265,328,363,364,365]. Recently, Unalan et al. (2023) and Liang et al. (2023) prepared (3D)-printed hydrogels with xanthan for burn wound healing [264,265]. Cakmak et al. developed a 3D-printed BC/polycaprolactone/gelatin/hydroxyapatite composite scaffold for bone tissue engineering [364]. Aki et al. also developed a 3D-printed PVA/hexagonal boron nitride/bacterial cellulose composite scaffold for bone tissue engineering [365]. 

## 5. Conclusions and Further Prospects

The present review summarizes strategies for the cost-effective production of important industrial exopolysaccharides such as xanthan and BC and demonstrates for the first time common approaches to their efficient production and to obtaining new functional materials for a wide range of applications, including wound healing, drug delivery, tissue engineering, environmental remediation, nanoelectronics, and 3D printing. Xanthan and BC are eco-friendly biopolymers with unique beneficial properties, such as biodegradability, biocompatibility, non-toxicity, a high degree of polymerization, the ability for gelation, renewable sourcing, and easy modification. Therefore, they have extensive commercial applications in biomedicine, food, pharmaceuticals, cosmetics, electronics, environmental remediation, the oil and gas industries, etc. The global xanthan market is developing rapidly due to applications in the food and agrochemical industries, cosmetics, driller fluid, and foam stabilizer segments. In addition, in recent years, the range of its application has expanded significantly, including biomedicine, environmental remediation, nanoelectronics, and 3D printing. Over the past few decades, BC production has also exponentially increased. The high cost of fermentation media is the limiting factor for BC and xanthan production. About 30% of the total cost accounts for the nutrient medium cost. Cost being a limiting factor in EPS production, many research investigations have been launched to use industrial wastes and by-products such as food and agro-industrial wastes, wastes from the sugar, dairy, alcohol, and biodiesel industries, and ABE fermentation. Statistical methods have been applied systematically to optimize xanthan and BC production parameters. Another strategy for efficient EPS production can be the co-cultivation of EPS-producing bacteria. Some publications specified the positive effect of the co-cultivation of EPS-producing bacteria on product yield. Biocatalytic techniques are promising for obtaining bacterial EPSs. Carbohydrate substrate conversion into a biopolymer can be improved by the activity of the enzymes involved in the synthesis and regulation of the pathways for EPS precursor biosynthesis. Cell immobilization can significantly improve the productivity and stability of biocatalysts. In recent years, valuable information has emerged on QS mechanisms in EPS biosynthesis. This area requires further study in terms of application for more efficient EPS production. Cell-free systems for EPS biosynthesis also show great development prospects. The cell-free systems may be a possible solution to the limitations faced by traditional EPS production, such as low yield and productivity, the production of byproducts and secondary metabolites, and high downstream processing costs. Another strategy for xanthan and BC cost-effective production is to isolate new bacterial strains from natural sources and create highly productive strains by genetic and metabolic engineering. With the help of genetic engineering, new strains with modified or introduced enzymatic activity can be created, which can expand the range of inexpensive substrates available for production and increase the degree of their transformation into EPSs. A growing number of studies is focused on the mechanisms involved in BC and xanthan biosynthesis, metabolic modeling, and genetic analysis applied to enable the improvement of its production on a large scale. Using genetic engineering for EPS production is a promising alternative to improve production on an industrial scale. However, it should be noted that despite numerous studies aimed at creating productive strains, none of them has so far led to the development of mutant strains which comply with the requirements of large-scale biotechnological production. Bacterial EPSs are characterized by a large number of functional groups which enable them to modify their molecules to give them new valuable properties. Therefore, a great number of EPS-based biocomposite materials have been obtained. The already-developed methodological approaches and the accumulated data on their modification will enable the creation of an even greater number of different functional materials with a wide range of applications in the future.

In conclusion, we would like to note that in our opinion, an integrated approach is required to further improve BC and xanthan production, taking into account all the strategies described in the review. Particular attention should be paid to genetic and metabolic engineering in order to obtain highly productive strains meeting the requirements of large-scale biotechnological production. Further research in the field of biocatalytic technologies and a deeper understanding of QS mechanisms in EPS biosynthesis are needed to produce BC and xanthan in the future. When solving the problems, there must be a clear understanding of the relationship between EPS production processes, their properties, and their possible uses for the targeted production of materials with specified properties.

## Figures and Tables

**Figure 2 ijms-24-14608-f002:**
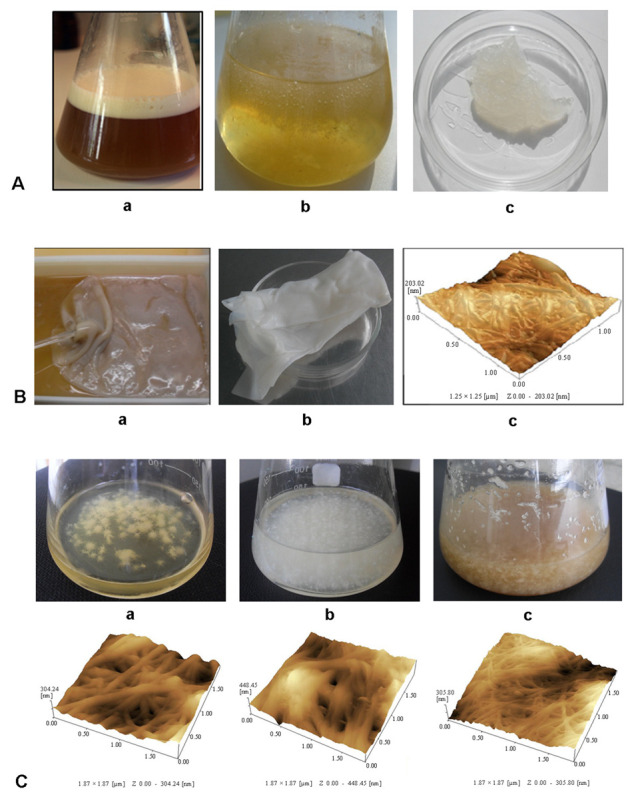
(**A**) Xanthan formed in agitated culture conditions, culture medium after fermentation (**a**), xanthan precipitation (**b**), precipitated xanthan (**c**); (**B**) BC gel film obtained in static conditions before (**a**) and after purification (**b**), and AFM image of the cellulose microfibrils (**c**); (**C**) BC produced by *K. sucrofermentans* B-11267 in dynamic conditions using HS medium (**a**), whey (**b**), and thin stillage (TS) (**c**), and AFM image of the cellulose microfibrils. Adapted from Refs. [2,46] (open access).

**Figure 3 ijms-24-14608-f003:**
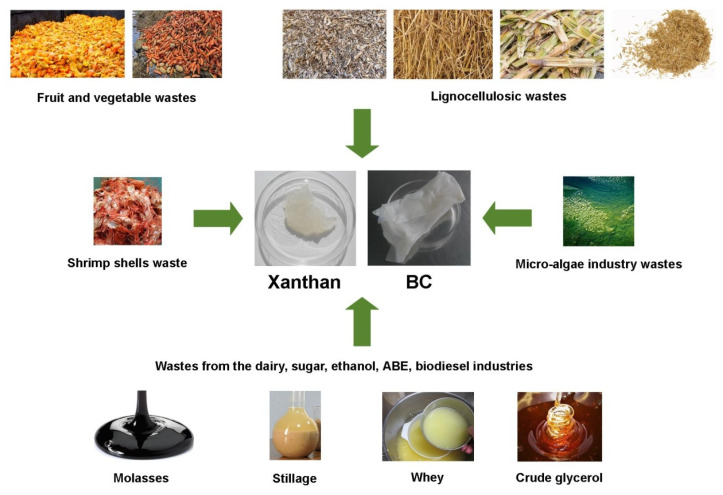
Schematic overview of BC and xanthan production from different food and agro-industrial wastes.

**Figure 4 ijms-24-14608-f004:**
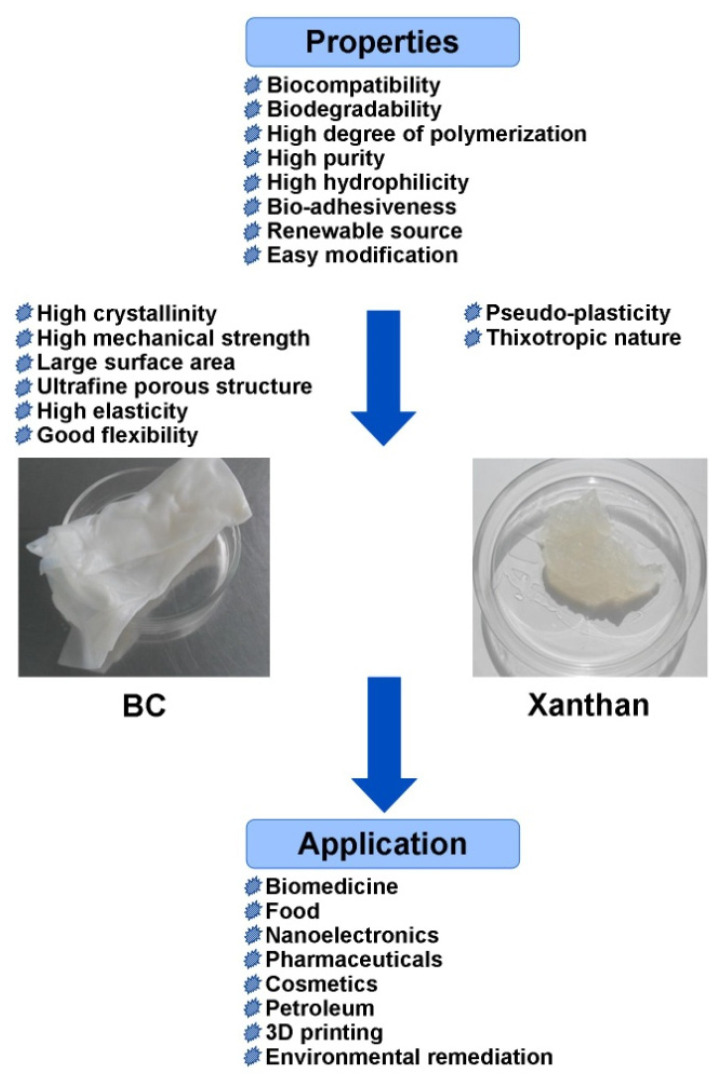
Properties and application of BC and xanthan.

**Table 1 ijms-24-14608-t001:** Food and agro-industrial wastes utilized as feedstocks for BC production by *Komagataeibacter* sp.

Bacterial Strains	Nutrient Source	Production Titer (g/L)	Reference
*Komagataeibacter* sp. PAP1	soya bean whey	4.10	[98]
*K. xylinus* CCM 3611	sour whey + cane sugar + black tea	18.5–23	[111]
*K. sucrofermentans* B-11267	cheese whey	5.45	[112]
	thin stillage	6.19	
	sugar beet molasses	2.9	[113]
*G. xylinus* PTCC 1734	cheese whey treated with β-galactosidase	3.55	[114]
*K. xylinus* K2G30	acid whey	2.99	[115]
*G. xylinus* FC01	molasses	0.572	[116]
*K. melomenusus* AV436T	grape pomace hydrolysate	1.24	[117]
*G. liquefaciens*	jaggery	17.79	[118]
*Komagataeibacter* sp. CCUG73629 and *Komagataeibacter* sp. CCUG73630	corncob and sugarcane bagasse	1.2–1.6	[119]
*K. sucrofermentans* DSM 15973	confectionery wastes	5.7	[120]
*K. sucrofermentans* DSM 15973	winery waste streams (grape pomace, stalks, wine lees)	8–11.6	[121]
*K. xylinus* BCRC12334	olive oil mill wastewater	0.65–5.33	[122]
*G. xylinus* ATCC 23770	wheat straw	8.3	[123]
*K. sucrofermentas* ATCC 700178	alkali lignin	3.2	[124]
*G. xylinus* CH001	durian shell	2.67	[125]
*K. rhaeticus*	cashew crop residues	2.3–6	[126]
*K. xylinus* DSM 6513	red grapes bagasse	0.548	[127]
*K. rhaeticus* M12	peer peal and pomace	10.94	[128]
*K. xylinus*	pineapple waste	3.82	[129]
*G. swingsii*	pineapple peel	2.8	[130]
*K. medellinensis* NBRC 3288	rotten banana waste	3.23	[131]
*K. europaeus* SGP37	sweet lime pulp	26.2	[132]
*G. xylinus* 0416 MARDI	date	5.8	[133]
*K. hansenii* GA2016	citrus peels	0.392	[134,135]
*G. xylinus* ATCC 700178	carob and haricot bean	1.8–3.2	[136]
*G. xylinus* 1.1812	sweet potato	11.35	[137]
*G. xylinus* BPR2001	sweet potato peel	2.3–4.2	[138]
*K. xylinus* TISTR 1011	yam bean	0.47	[139]
*G. xylinus* BPR2001	okara	2.3	[140]
*G. aceti* ATCC 23770	konjac powder	2.12	[141]
*K. rhaeticus* K15	kitchen waste	4.76	[142]
*Komagataeibacter* sp. PAP1	noodle wastewater	11.76	[143]
*G. hansenii* UAC09	coffee cherry husk	8.2	[144]
*G. xylinus* KCCM 41431 (ATCC 11142)	pure glycerol crude glycerol	3.4 2.93	[145]

**Table 2 ijms-24-14608-t002:** Food and agro-industrial wastes utilized as feedstocks for xanthan production.

Bacterial Strains	Nutrient Source	Production Titer (g/L)	Reference
*X. campestris*	orange peels hydrolysate	30.19	[146]
*X. campestris*	carrot peels extract	40.88	[104]
	pumpkin peels extract	31.4	
*X. campestris* NCIM 2954	tapioca pulp hydrolysate	7.1	[147]
*X. campestris* NRRL B-1459	date juice	43.35	[148]
*X. campestris* pv. *vesicatoria*	waste bread hydrolysate	14.3	[149]
*X. campestris* LRELP-1	kitchen waste hydrolysate	4.09–6.46	[150]
*X. campestris* NCIM 2961	jackfruit seed powder	51.62	[151]
*Xanthomonas* sp. 629	hemicellulose fractions from the alkaline extraction of corncob	8.37	[152]
*X. campestris* pv. *campestris* 1866	cocoa husks	4.48	[153]
*X. campestris* pv. *campestris* 1867	acid hydrolyzed broomcorn stem	3.89	[91]
*X. campestris*		8.9	
*X. campestris* CCTCC M2015714	glycerol	11.0	[154]
*X. campestris* Xp 3–1	crude glycerol	7.67	[155]
*X. campestris* pv. *mangiferaeindicae* 2103	crude glycerol	5.59	[156]
*X. citri*/NIGEB-386	cheese whey	22.7	[157]
*X. campestris*	cheese whey	16.4	[158]
*X. pelargonii*		12.8	
*X. campestris* M 28	molasses	28	[159]
*X. campestris* ATCC 13951	wastewaters during the washing operations of the crusher	4.004	[160]
	wastewaters during the washing operations of the press	7.596	
	wastewaters during the washing operations of the tanks after clarification of must	10.67	
	wastewaters during the washing operations of the fermentation	7.488	
*X. campestris* pv. *manihotis* 1182	2% concentration of the shrimp shell used in the aqueous extract	2.64	[161]
*X. campestris* pv. *campestris* 254		2.60	
*X. campestris* pv. *campestris* 629		1.95	

## Data Availability

No new data were created or analyzed in this study. Data sharing is not applicable to this article.

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
