# Peer review of "Production of Bacterial Exopolysaccharides: Xanthan and Bacterial Cellulose"

_ijms, 2023, doi:10.3390/ijms241914608_

Round 1
Reviewer 1 Report
This review describes the material characteristics, production, industrial applications and biosynthesis of the two commercially significant bacterial exopolysaccharides cellulose and xanthan.
The review is in general well written. Major comments include the organization of the descriptions at several occasions whereby description of the parts concerning cellulose and xanthan, after introductory sentences, should be clearly separated. For example, in part two, one can start with the properties of the cellulose macromolecule, as indicated in Figure 1A and then go to the more complex xanthan. This is also indicated in Figure1, where display of the cellulose chemical subunits as shown in Figure 1A are related to Figure 1D, but for consistency 1D should be 1C.
Another major point is that some sentences are not really informative. This concerns, for example, but is not limited to: l. 49, l.629, l639.
Also, references to major original findings with respect to cellulose biosynthesis and regulation of cellulose production are relevant to provide. This includes, but is not limited to, original findings by the Benziman group on regulation of cellulose biosynthesis by cyclic di-GMP.
One or two additional figures would make the review more attractive.
Other comments:
Introduction: Can give first a general picture of the subject, but should highlight at the end of each paragraph what are the specific features with respect to xanthan and cellulose basically arguing why these exopolysaccharides are exceptional and worthwhile to describe from different aspects.
l. 241: does this growth medium have a name?
l. 272: cheap?, inexpensive
l. 320: list all relevant resources
l. 490: crystallinity of what?, the cellulose?
l. 536: point 3.2.3 Genetic and metabolic engineering, more concrete examples can be given on several occasions.
l.567: …which can limit…
l. 585: again, this depends how the mutants have been constructed.
l. 809: Should contain the authors’ own innovative opinion how one can in the best way tackle specific questions.
quite ok
Author Response
Thank you very much for all of your detailed comments and suggestions. We found them quite useful as we approached our revision. The authors completely agree with reviewers comment.
- Major comments include the organization of the descriptions at several occasions whereby description of the parts concerning cellulose and xanthan, after introductory sentences, should be clearly separated. For example, in part two, one can start with the properties of the cellulose macromolecule, as indicated in Figure 1A and then go to the more complex xanthan. This is also indicated in Figure1, where display of the cellulose chemical subunits as shown in Figure 1A are related to Figure 1D, but for consistency 1D should be 1C.
RESPONSE: Thank you very much for your valuable suggestions. We have separated the parts concerning cellulose and xanthan in Chapter 2 and swapped the positions of the Figure 1D and Figure 1C.
- Another major point is that some sentences are not really informative. This concerns, for example, but is not limited to: l. 49, l.629, l639.
RESPONSE: Thank you very much for your valuable suggestions. We have added information to some sentences. For example:
- 49 «Recently, several reviews on EPS produced by lactic acid bacteria have been reported [24-27]. For example, the review by Jurášková et al. (2022) discussed and summarized the latest advances on biosynthesis, structure, and properties of EPS derived from Lactobacillus, Leuconostoc, Streptococcus, Lactococcus, Lactiplantibacillus, Limosilactobacillus, Weissella genera [24]».
- 639 «Recently, the review by Horue et al. (2023) provides the information on BC-based materials as dressings for wound healing [53]. Authors reported the main characteristics of different BC structures such as films, membranes, fibrous, etc., as well as recent advances in BC-based composites».
- Also, references to major original findings with respect to cellulose biosynthesis and regulation of cellulose production are relevant to provide. This includes, but is not limited to, original findings by the Benziman group on regulation of cellulose biosynthesis by cyclic di-GMP.
RESPONSE: Thank you very much for your valuable suggestion. We have added references to major original findings. «One of the well-characterized mechanisms regulating BC biosynthesis is the allosteric activation of BcsA with a cyclic di-GMP (c-di-GMP) molecule, a universal bacterial second messenger discovered by Moshe Benziman and his group in 1987 after many years of studying the mechanism and regulation of BC biosynthesis in the bacterium Acetobacter xylinum (K. xylinus) [85,86]».
- One or two additional figures would make the review more attractive.
RESPONSE: Thank you very much for your recommendation. We have added Figure 5.
- Introduction: Can give first a general picture of the subject, but should highlight at the end of each paragraph what are the specific features with respect to xanthan and cellulose basically arguing why these exopolysaccharides are exceptional and worthwhile to describe from different aspects.
RESPONSE: Thank you very much for your valuable suggestion. We have added conclusions at the end of paragraphs and highlighted what are the specific features with respect to xanthan and cellulose.
«In recent years, a large number of new strains of EPS-producing bacteria have been isolated, among them the Gram-negative bacteria Komagataeibacter sp. and Xanthomonas sp., the main producers of BC and xanthan, deserve special attention, since their metabolism is well studied, and many complete genome sequences were obtained, which is the basis for obtaining highly productive strains using genetic and metabolic engineering [2]».
«Among bacterial EPS, the homopolysaccharide BC and the heteropolysaccharide xanthan rank high. They exhibit many unique properties. For example, BC is a 3D nanostructured material with high crystallinity and a large surface area, and xanthan, along with other valuable properties, has pseudo-plasticity, thixotropic nature, and is resistant to various environmental factors [2]. Over the last years, there have been obtained a great number of xanthan and BC-based biocomposite materials with additional valuable properties, including antimicrobial activity, antioxidant activity, electromagnetic properties, catalytic activity, and others [5,10-12].»
- l. 241: does this growth medium have a name?
RESPONSE: This is the general name for different culture media containing sucrose and salts.
- l. 272: cheap?, inexpensive
RESPONSE: The word «cheap» is replaced by «inexpensive».
- l. 320: list all relevant resources
RESPONSE: This paragraph provides the information on such dairy, sugar and alcohol industrial wastes as whey, molasses and stillage.
- l. 490: crystallinity of what?, the cellulose?
RESPONSE: The text was corrected. «The presence of hyaluronic acid improved water holding capacity of the composites resulting in a decrease in BC crystallinity».
- l. 536: point 3.2.3 Genetic and metabolic engineering, more concrete examples can be given on several occasions.
RESPONSE: We have added examples. «Jang et al. demonstrated that K. xylinum strain overexpressing the E. coli glucose 6-phosphate isomerase gene produced 3.15 g/L of BC, which was 115.8% higher as compared to 1.46 g/L obtained from the control strain [243]».
- l. 567: …which can limit…
RESPONSE: The text was changed. «Most genetic studies were conducted by transposon mutagenesis, which can limit the interpretability of the results due to polar effects and other artefacts [87]».
- l. 585: again, this depends how the mutants have been constructed.
RESPONSE: The authors agree with this opinion.
- l. 809: Should contain the authors’ own innovative opinion how one can in the best way tackle specific questions.
RESPONSE: Thank you very much for your recommendation. We have made changes to Chapter 5 and added information.
«In conclusion, we would like to note that in our opinion, an integrated approach is required to further improve BC and xanthan production taking into account all the strategies described in the review. Particular attention should be paid to genetic and metabolic engineering in order to obtain highly productive strains meeting the requirements of large-scale biotechnological production. Further research in the field of biocatalytic technologies and a deeper understanding of QS mechanisms in EPS biosynthesis are needed to produce BC and xanthan in future. When solving the problems, there must be a clear understanding of the relationship between EPS production processes, their properties and possible use for the targeted production of materials with specified properties».

Reviewer 2 Report
The work is well prepared. This is a comprehensive review presenting the reader with the strategies for the cost-effective production of xanthan and bacterial cellulose. The first chapter should give the structure of the work and describe what is given in each chapter. The work may lack a description related to the apparatus used in the BC and xanthan production process (this may be an idea for another work).
Additional comments
The presented work is connected with the production of bacterial exopolysaccharides (xanthan and bacterial cellulose). These polymers have unique properties that allow their use in biomedicine, foodstuffs, textiles, etc. The main aim of the paper is to demonstrate strategies for cost-effective production of these important biomacromolecules. In my opinion, this paper is crucial for researchers because it showcases the state-of-the-art of the production processes for xanthan and bacterial cellulose. The presented work summarizes 372 papers and contains valuable information related to the effective production of eco-friendly polymers.
In general, this paper is well-prepared and is divided into five chapters. I suggest that the authors consider the following comments to improve their work:
- The first chapter should provide an overview of the work's structure and describe what is covered in each chapter. It might also be beneficial to include a section on the equipment used in the production processes of bacterial cellulose and xanthan. This could potentially be the subject of future research.
- I recommend adding information about the physical and chemical properties of the described polymers. This information could be incorporated into Chapter 2.
- It would be beneficial to include information about the cost of the production process of these polymers in the text.
- The production methods of these polymers should be described in more detail. For instance, the description of both static and agitated conditions should be provided (see Fig. 2).
- Information related to bioprocessing and the equipment used in the production of these polymers should be included.
- In general, the economic aspects of the production process for these biopolymers should be addressed. There is a lack of information regarding the cost of producing, for example, 1 kg of polymer.
Minor language correction is required.
Author Response
Thank you very much for all of your comments and suggestions. We found them quite useful as we approached our revision. The authors completely agree with reviewers comment.
- The first chapter should provide an overview of the work's structure and describe what is covered in each chapter. It might also be beneficial to include a section on the equipment used in the production processes of bacterial cellulose and xanthan. This could potentially be the subject of future research.
RESPONSE: Thank you very much for your recommendation. Manuscript corrections have been made.
«Chapter 1 presents a general idea of the subject highlighting the specific features of xanthan and cellulose compared to other bacterial EPSs. Chapter 2 provides the information on properties, biosynthesis, as well as BC and xantan producers. Chapter 3 summarizes the strategies for xanthan and BC cost-effective production including their production from wastes, EPS-producing bacteria co-cultivation, biocatalytic technologies, and genetic and metabolic engineering. Chapter 4 introduces the recent advances in obtaining new functional materials for a wide range of applications including wound healing, drug delivery, tissue engineering, environmental remediation, nanoelectronics, and 3D bioprinting. At the end of the review, we recommend further studies and investigation for highly efficient BC and xanthan production, and obtaining new functional materials based on them».
We have added some information on the equipment used in the production processes of bacterial cellulose and xanthan into Chapter 3».
- I recommend adding information about the physical and chemical properties of the described polymers. This information could be incorporated into Chapter 2.
RESPONSE: We have added some information on the physical and chemical properties of the described polymers into Chapter 2.
«This unique three-dimensional network structure of BC is responsible for most of its properties such as high tensile strength, high degree of polymerization and crystallinity (up to 90%), superior mechanical properties (Young’s modulus about 15–35 GPa and tensile strength of 200–300 MPa), large surface area (>150 m2/g), high elasticity, and water retention [37,54-58]»
«Xanthan in contrast to BС is a water-soluble bacterial EPS and has high solubility in both cold and hot water». «Xanthan has pseudoplastic nature; that is, the viscosity inversely changes with shear rate of a xanthan solution». «Xanthan has a high tolerance to deviations in the pH range of 2–12 and high resistance to temperature changes».
- It would be beneficial to include information about the cost of the production process of these polymers in the text.
RESPONSE: We have added information on the cost of the production process of these polymers.
- The production methods of these polymers should be described in more detail. For instance, the description of both static and agitated conditions should be provided (see Fig. 2).
RESPONSE: We described the production methods of these polymers in more detail.
The following information has been added to the manuscript:
«The optimum temperature and pH for xanthan production is 28–30°C and pH 7–8, respectively. Various types of bioreactors have been used for xanthan production, but mechanically stirred bioreactors are the most common. In stirred reactors the rate of oxygen mass transfer is affected by air flow rate and a stirrer speed. The air flow rate is usually maintained at a constant level, typically 1 L/L min. The stirrer speed usually varies during culture from lower values (200–300 rpm) at the beginning of the fermentation to higher values (400–600 rpm) later on».
«The method selection depends on final BC applications as well as the required morphological, mechanical and physicochemical characteristics [100]. The static cultivation method is a traditional approach for BC production, and it has become widespread. According to the method, bacteria are grown in the containers with a growth medium, usually for 7–14 days at 28–30 °C and pH of 4–7. A BC gel film forms on the medium surface, its size depending on the surface area of the medium (Fig. 2Ba). After purification with sodium hydroxide by heating and further washing deionized water until pH became neutral, the film becomes colorless and transparent (Fig. 2Bb). Figure 2Bc shows BC micromorphology, which exhibits a nanoporous three-dimensional network structure with a random arrangement of ribbon-shaped fibrils. High cost and low rate of production are the two main problems in static culture systems. However, BC produced using static condition has more advantages in biomedical fields due to higher crystallinity, linear structure, good flexibility, and elasticity. Agitated cultivation is the preferred method for BC industrial production. Less space is required in agitated cultivation compared to the static condition, and aeration rate is higher during BC production. Different types of bioreactors are utilized for this purpose including stirred tank bioreactors, airlift bioreactors, rotating disk reactors, rotary biofilm contactor reactors, and reactors with silicone membranes [100]. BC obtained under dynamic conditions has a high water-holding capacity, but usually lower crystallinity».
- Information related to bioprocessing and the equipment used in the production of these polymers should be included.
RESPONSE: We included information related to bioprocessing and the equipment used in the production of these polymers.
- In general, the economic aspects of the production process for these biopolymers should be addressed. There is a lack of information regarding the cost of producing, for example, 1 kg of polymer.
RESPONSE: We have added information on the cost of the production process of these polymers. «BC market price is about US$25/kg for the packaged final product [45]. The market price of xanthan is next lower order, and amounts to about US$1500-4000/ton [47]».
